# Dynamic Network Reconfiguration for Entropy Maximization using Deep Reinforcement Learning

**Christoffel Doorman[1], Victor-Alexandru Darvariu[1,2], Stephen Hailes[1], Mirco Musolesi[1,2,3]**
[1]University College London    [2]The Alan Turing Institute    [3]University of Bologna
{christoffel.doorman.20, v.darvariu, s.hailes, m.musolesi}@ucl.ac.uk

## Abstract

A key problem in network theory is how to reconfigure a graph in order to optimize a quantifiable objective. Given the ubiquity of networked systems, such work has broad practical applications in a variety of situations, ranging from drug and material design to telecommunications. The large decision space of possible reconfigurations, however, makes this problem computationally intensive. In this paper, we cast the problem of network rewiring for optimizing a specified structural property as a Markov Decision Process (MDP), in which a decision-maker is given a budget of modifications that are performed sequentially. We then propose a general approach based on the Deep Q-Network (DQN) algorithm and graph neural networks (GNNs) that can efficiently learn strategies for rewiring networks. We then discuss a cybersecurity case study, i.e., an application to the computer network reconfiguration problem for intrusion protection. In a typical scenario, an attacker might have a (partial) map of the system they plan to penetrate; if the network is effectively "scrambled", they would not be able to navigate it since their prior knowledge would become obsolete. This can be viewed as an entropy maximization problem, in which the goal is to increase the *surprise* of the network. Indeed, entropy acts as a proxy measurement of the difficulty of navigating the network topology. We demonstrate the general ability of the proposed method to obtain better entropy gains than random rewiring on synthetic and real-world graphs while being computationally inexpensive, as well as being able to generalize to larger graphs than those seen during training. Simulations of attack scenarios confirm the effectiveness of the learned rewiring strategies.

## 1 Introduction

A key problem in network theory is how to rewire a graph in order to optimize a given quantifiable objective. Addressing this problem might have applications in several domains, given the fact several systems of practical interest can be represented as graphs [23, 24, 29, 50, 51]. A large body of literature studies how to construct and design networks in order to optimize some quantifiable goal, such as robustness in supply chain and wireless sensor networks [40, 54] or ADME properties of molecules [18, 39]. Given the intractable number of distinct configurations of even relatively small networks, optimizing these structural and topological properties is generally a non-trivial task that has been approached from various angles in graph theory [14, 17] and also studied from heuristic perspectives [21, 35]. Exact solutions are too computationally expensive and heuristic methods are generally sub-optimal and do not generalize well to unseen instances.

The adoption of graph neural networks (GNNs) [41] and deep reinforcement learning (RL) [36] techniques have led to promising approaches to the problem of optimizing graph processes or structure [13, 15, 30]. A fundamental structural modification is *rewiring*, in which edges (e.g., links in a computer network) are reconfigured such that the topology is changed while their total number remains constant. The problem of rewiring to optimize a structural property has not been studied in the literature.

C. Doorman et al., Dynamic Network Reconfiguration for Entropy Maximization using Deep Reinforcement Learning. *Proceedings of the First Learning on Graphs Conference (LoG 2022)*, PMLR 198, Virtual Event, December 9–12, 2022.

In this paper, we present a solution to the network rewiring problem for optimizing a specified structural property. We formulate this task as a Markov Decision Process (MDP), in which a decision-maker is given a budget of rewiring operations that are performed sequentially. We then propose an approach based on the Deep Q-Network (DQN) algorithm and GNNs that can efficiently learn strategies for rewiring networks. We evaluate the method by means of a realistic cybersecurity case study. In particular, we assume a scenario in which an attacker has entered a computer network and aims to reach a particular node of interest. We also assume that the attacker has partial knowledge of the underlying graph topology, which is used to reach a given target inside the network. The goal is to learn a rewiring process for modifying the structure of the graph so as to disrupt the capability of the attacker to reach its target, all the while keeping the network operational. This can be seen as an example of *moving target defense* (MTD) [7]. We frame the solution as an entropy maximization problem, in which the goal is to increase the *surprise* of the network in order to disrupt the navigation of the attacker inside it. Indeed, entropy acts as proxy measurement of the difficulty of this task, with an increase in the entropy of the graph corresponding to a more challenging navigation task. In particular, we consider two measures of network entropy – namely Shannon entropy and Maximal Entropy Random Walk (MERW), and we compare their effectiveness.

More specifically, the contributions of this paper can be summarized as follows:

- We formulate the problem of graph rewiring so as to maximize a global structural property as an MDP, in which a central decision-maker is given a certain budget of rewiring operations that are performed sequentially. We formulate an approach that combines GNN architectures and the DQN algorithm to learn an optimal set of rewiring actions by trial-and-error;

- We present an extensive case study of the proposed approach in the context of defense against network intrusion by an attacker. We show that our method is able to obtain better gains in entropy than random rewiring, while scaling to larger networks than a local greedy search, and generalizing to larger out-of-distribution graphs in some cases. Furthermore, we demonstrate the effectiveness of this approach by simulating the movement of an attacker in the network, finding that indeed the applied modifications increase the difficulty for the attacker to reach its targets in both synthetic and real-world graph topologies.

## 2 Related work

**RL for graph reconfiguration.** Recently, an increasing amount of research has been conducted on the use of reinforcement learning in graph reconfiguration. In particular, in [13] a solution based on reinforcement learning for modifying graphs with the aim of attacking both node and graph classification is presented. In addition, the authors briefly introduce a defense method using adversarial training and edge removal, which decreases their proposed classifier attack rate slightly by 1%. This defense strategy is however only effective on the attack strategy it is trained on and does not generalize. Instead, the authors of [34] use a reinforcement learning approach to learn an attack strategy for neural network classifiers of graph topologies based on edge rewiring, and show that they are able to achieve misclassification with changes that are less noticeable compared to edge and vertex removal and addition. Our paper focuses on a different problem that does not involve classification tasks, but the maximization of a given network objective function. In [15] reinforcement learning techniques are applied to the problem of optimizing the robustness of a graph by means of graph construction; the authors show that their proposed method is able to outperform existing techniques and generalize to different graphs. In the present work, we optimize a global structural property through rewiring instead of constructing a graph through edge addition.

**Graph robustness and attacks.** A related research area is the optimization of *graph robustness* [37], which denotes the capacity of a graph to withstand targeted attacks and random failures. [42] demonstrates how small changes in complex networks such as an electricity system or the Internet can improve their robustness against malicious attacks. [5] investigates several heuristic reconfiguration techniques that aim to improve graph robustness without substantially modifying the network structure, and find that preferential rewiring is superior to random rewiring. The authors of [10] extend this study to a framework that can accommodate multiple rewiring strategies and objectives. Several works have used information-based complexity metrics in the context of network defense or attack strategies: [27] proposes a network security metric to assess network vulnerability by measuring the Kolmogorov complexity of effective attack paths. The underlying reasoning is that the more complex attack paths have to be in order to harm a network, the less vulnerable a network is to external attacks.

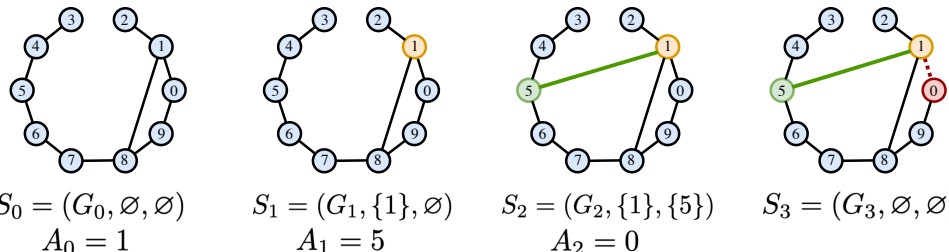

**Figure 1:** Illustrative example of the MDP timesteps comprising a single rewiring operation. The agent observes an initial state $S_0 = (G_0, \varnothing, \varnothing)$ (first panel), from which it then selects a base node $v_1 = \{1\}$ that will be rewired (second panel). Given the new state that contains the initial graph and the selected base node, the agent selects a target node $v_2 = \{5\}$ to which an edge will be added (third panel). Finally, a third node $v_3 = \{0\}$ is selected from the neighborhood of $v_1 = \{1\}$ and the corresponding edge is removed (last panel). After a *sequence* of $b$ rewiring operations, the agent will receive a reward proportional to the improvement in the objective function $\mathcal{F}$.

Furthermore, [25] investigates the vulnerability of complex networks, finding that attacks based on edge and vertex removal are substantially more effective when the network properties are recomputed after each attack.

**Cybersecurity and network defense.** In the last decade and in recent years in particular, a drastic surge in cyberattacks on governmental and industrial organizations has exposed the imminent vulnerability of global society to cyberthreats [43]. The targeted digital systems are generally structured as a network in which entities in the system communicate and share resources among each other. Typically, attackers seek to gain unauthorized access to the underlying network through an entry point and search for highly valuable nodes in order to infect these digital systems with malicious software such as viruses, ransomware and spyware [2], enabling them to extract sensitive information or control the functioning of the network [26]. Moving target defense (MTD) is a cybersecurity defense technique by which a network and the underlying software are dynamically changed to counteract attack strategies [3, 7, 8, 44, 52]. Most existing MTD techniques involve NP-hard problems, and approximate or heuristic solutions are often impractical [7]. We note that while most studies are applied to specific software architectures, which prevent them from being applied effectively to large scale deployments, in this work we focus on modeling this problem from an abstract, infrastructure-agnostic perspective.

## 3 Graph rewiring as an MDP

### 3.1 Problem statement

We define a graph (network) as $G = (\mathcal{V}, \mathcal{E})$, where $\mathcal{V} = \{v_1, ..., v_n\}$ is the set of $n = |\mathcal{V}|$ vertices (nodes) and $\mathcal{E} = \{e_1, ..., e_m\}$ is the set of $m = |\mathcal{E}|$ edges (links). A *rewiring* operation $\gamma(G, v_i, v_j, v_k)$ transforms the graph $G$ by adding the non-edge $(v_i, v_j)$ and removing the existing edge $(v_i, v_k)$; we denote the set of all such operations by $\Gamma$. Given a budget $b \propto m$ of rewiring operations, and a global objective function $\mathcal{F}(G)$ to be maximized, the goal is to find the set of unique rewiring operations out of $\Gamma^b$ such that the resulting graph $G'$ maximizes $\mathcal{F}(G')$. Since the size of the set of possible rewirings grows rapidly with the graph size, we cast this problem as a sequential decision-making process, which is detailed below.

### 3.2 MDP framework

We let every rewiring operation consist of three sub-steps: 1) base node selection; 2) node selection for edge addition; and 3) node selection for edge removal. We precede the edge removal step by edge addition to suppress potential disconnections of the graph. The rewiring procedure is illustrated in Figure 1. For reducing the size of the decision space, we model each sub-step of the rewiring operation as a separate timestep in the MDP itself. Its elements are defined as:

**State.** The state $S_t$ is the tuple $S_t = (G_t, a_1, a_2)$, containing the graph $G_t = (\mathcal{V}, \mathcal{E}_t)$, the chosen base node $a_1$, and the chosen addition node $a_2$. The base node and addition node may be null ($\varnothing$) depending on the rewiring operation sub-step.

**Actions.** We specify three distinct action spaces $\mathcal{A}_{\hat{t}}(S_t)$, where $\hat{t} := (t \mod 3)$ denotes the sub-step within a rewiring operation. Letting the degree of node $v$ be $k_v$, they are defined as:

$$\mathcal{A}_0\Big(S_t = \big((\mathcal{V}, \mathcal{E}_t), \varnothing, \varnothing\big)\Big) = \big\{v \in \mathcal{V} \mid 0 < k_v < |\mathcal{V}| - 1\big\}, \tag{1}$$

$$\mathcal{A}_1\Big(S_t = \big((\mathcal{V}, \mathcal{E}_t), a_1, \varnothing\big)\Big) = \big\{v \in \mathcal{V} \mid (a_1, v) \notin \mathcal{E}_t\big\}, \tag{2}$$

$$\mathcal{A}_2\Big(S_t = \big((\mathcal{V}, \mathcal{E}_t), a_1, a_2\big)\Big) = \big\{v \in \mathcal{V} \mid (a_1, v) \in \mathcal{E}_t \setminus (a_1, a_2)\big\}. \tag{3}$$

**Transitions.** Transitions are deterministic; the model $P(S_t = s' | S_{t-1} = s, A_{t-1} = a_{t-1})$ transitions to state $S'$ with probability 1, where:

$$S' = \begin{cases} \big((\mathcal{V}, \mathcal{E}_{t-1}), a_1, \varnothing\big) & \text{if } 3 \mid t+2 \qquad \textit{mark base node,} \\ \big((\mathcal{V}, \mathcal{E}_{t-1} \cup (a_1, a_2)), a_1, a_2\big) & \text{if } 3 \mid t \qquad \textit{mark addition node \& add edge,} \\ \big((\mathcal{V}, \mathcal{E}_{t-1} \setminus (a_1, a_3)), \varnothing, \varnothing\big) & \text{if } 3 \mid t+1 \qquad \textit{remove edge \& reset marked nodes.} \end{cases} \tag{4}$$

**Rewards.** The reward signal $R_t$ is proportional to the difference in the value of the objective function $\mathcal{F}$ before and after the graph reconfiguration. Furthermore, a key operational constraint in the domain we consider is that the network remains connected after the rewiring operations. Instead of running connectivity algorithms at every timestep to determine if a potential removed edge disconnects the graph, we encourage maintaining connectivity by giving a penalty $\bar{r} < 0$ at the end of the episode if the graph becomes disconnected. All rewards and penalties are provided at the final timestep $T$, and no intermediate rewards are given. This enables the flexibility to discover long-term strategies that maximize the total cumulative reward of a sequence of reconfigurations rather than a single-step rewiring operation, even if the graph is disconnected during intermediate steps. Concretely, given an initial graph $G_0 = (\mathcal{V}, \mathcal{E}_0)$, we define the reward function at timestep $t$ as:

$$R_t = \begin{cases} c_{\mathcal{F}} \cdot \big(\mathcal{F}(G_t) - \mathcal{F}(G_0)\big) & \text{if } t = T \wedge c(G) = 1, \\ \bar{r} & \text{if } t = T \wedge c(G) \geq 2, \\ 0 & \text{otherwise,} \end{cases} \tag{5}$$

where $c(G)$ denotes the number of connected components of $G$, and $\bar{r} < 0$ is the disconnection penalty. As the different objective functions may act on different scales, we use a reward scaling $c_{\mathcal{F}}$, which we empirically establish for every objective function $\mathcal{F}$.

## 4 Reinforcement learning representation and parametrization

In this section, we extend the graph representation and value function approximation parametrizations proposed in past work [13, 15] for the problem of graph rewiring.

### 4.1 Graph representation

As the state and action spaces in network reconfiguration quickly become intractable for a sequence of rewiring operations, we require a graph representation that generalizes over similar states and actions. To this end, we use a GNN architecture that is based on a mean field inference method [47]. More specifically, we use a variant of the *structure2vec* [12] embedding method to represent every node $v_i \in \mathcal{V}$ in a graph $G = (\mathcal{V}, \mathcal{E})$ by an embedding vector $\mu_i$. This embedding vector is constructed in an iterative process by linearly transforming feature vectors $x_i$ with a set of weights $\{\theta^{(1)}, \theta^{(2)}\}$, aggregating the $x_i$ with the feature vectors of neighboring nodes $v_j \in \mathcal{N}_i$, then applying the nonlinear Rectified Linear Unit (ReLU) activation function. Hence, at every step $l \in (1, 2, \ldots, L)$, embedding vectors are updated according to:

$$\mu_i^{(l+1)} = \text{ReLU}\left(\theta^{(1)} x_i + \theta^{(2)} \sum_{j \in \mathcal{N}_i} \mu_j^{(l)}\right), \tag{6}$$

where all embedding vectors are initialized as $\mu_i^{(0)} = \mathbf{0}$. After $L$ iterations of feature aggregation, we obtain the node embedding vectors $\mu_i \equiv \mu_i^{(L)}$. By summing the embedding vectors of nodes in a

graph $G$, we obtain its permutation-invariant embedding: $\mu(G) = \sum_{i \in \mathcal{V}} \mu_i$. These invariant graph embeddings represent part of the state that the RL agent observes. Aside from permutation invariance, such embeddings allow learned models to be applied to graphs of different sizes, potentially larger than those seen during training.

## 4.2 Value function approximation

Due to the intractable size of the state-action space in graph reconfiguration tasks, we make use of neural networks to learn approximations of the state-action values $Q(s, a)$ [48]. More specifically, as the action spaces defined in Equation (1) are discrete, we use the DQN algorithm [36] to update the state-action values as follows:

$$Q(s,a) \leftarrow Q(s,a) + \alpha \left[ r + \gamma \max_{a' \in \mathcal{A}} Q(s', a') - Q(s, a) \right]. \tag{7}$$

The DQN algorithm uses an experience replay buffer [33] from which it samples previously observed transitions $(s, a, r, s')$, and periodically synchronizes a target network with the parameters of the Q-network. The target network is used in the computation of the learning target for estimating the Q-value of the best action in the next timestep, making the learning more stable as the parameters are kept fixed between updates. We use three separate MLP parametrizations of the Q-function, each corresponding to one of the three sub-steps of the rewiring procedure:

$$Q_1\big(S_t = (G_t, \varnothing, \varnothing), A_t\big) = \theta^{(3)} \text{ReLU}\left( \theta^{(4)} \left[ \mu_{A_t} \oplus \mu(G_t) \right] \right), \tag{8a}$$

$$Q_2\big(S_t = (G_t, a_1, \varnothing), A_t\big) = \theta^{(5)} \text{ReLU}\left( \theta^{(6)} \left[ \mu_{a_1} \oplus \mu_{A_t} \oplus \mu(G_t) \right] \right), \tag{8b}$$

$$Q_3\big(S_t = (G_t, a_1, a_2), A_t\big) = \theta^{(7)} \text{ReLU}\left( \theta^{(8)} \left[ \mu_{a_1} \oplus \mu_{a_2} \oplus \mu_{A_t} \oplus \mu(G_t) \right] \right), \tag{8c}$$

where $\oplus$ denotes concatenation. We highlight that, since the underlying structure2vec parameters shown in Equation (6) are shared, the combined set of the learnable parameters in our model is $\Theta = \{\theta^{(i)}\}_{i=1}^{8}$. During validation and test time, we derive a greedy policy from the above learned Q-functions as $\arg \max_{a \in \mathcal{A}_t} Q(s, a)$. During training, however, we use a linearly decaying $\epsilon$-greedy behavioral policy. We refer the reader to Appendix A for a detailed description of our implementation.

# 5 Case study: network reconfiguration for intrusion defense

In this section, we detail the specifics of our intrusion defense application scenario. We first present the definition of the objective functions we leverage, which act as proxy metrics for the difficulty of navigating the graph. Secondly, we detail the procedure we use for simulating attacker behavior during an intrusion, which will allow us to compare the pre- and post-rewiring costs of traversal.

## 5.1 Objective functions for network obfuscation

Our goal is to reconfigure the network so as to deter an attacker with partial knowledge of the network topology. Equivalently, we seek to modify the network so as to increase the *surprise* of the network and render this prior knowledge obsolete, while keep the network operational. A natural formalization of surprise is the concept of entropy, which measures the quantity of information encoded in a graph or, equivalently, its complexity.

As measures of entropy, we investigate two graph quantities that are invariant to permutations in representation: the *Shannon entropy* of the degree distribution [45] and the *Maximum Entropy Random Walk (MERW)* [6] calculated from the spectrum of the adjacency matrix. The former captures the idea that graphs with heterogeneous degrees are less predictable than regular graphs, while the latter is related to random walks on the network. Whereas generic random walks generally do not maximize entropy [16], MERW uses a specific choice of transition probabilities that ensures every trajectory of fixed length is equiprobable, resulting in a maximal global entropy in the limit of infinite trajectory length. Although the local transition probabilities depend on the global structure of the graph, the generating process is local [6]. More formally, the two objective functions are formulated as follows: the Shannon entropy is defined as $\mathcal{F}_{\text{Shannon}}(G) = -\sum_{k=1}^{n-1} q(k) \log_2 q(k)$, where $q(k)$ is the degree distribution; MERW is defined as $\mathcal{F}_{\text{MERW}}(G) = \ln \lambda$, where $\lambda$ is the largest eigenvalue of

**Figure 2:** Illustrative example of the evaluation process for a network reconfiguration. (i) The graph is rewired by our approach, removing and adding the highlighted edges respectively. (ii) The leftmost nodes in the graph become unreachable by the attacker from the entry point marked E, and hence a path to them must be rediscovered by exploring the graph. (iii) To reach the nodes, the attacker pays a cost of 1 and 2 respectively for "unlocking" the previously unseen links along the highlighted paths. The total cost induced by the rewiring strategy is $\mathcal{C}_{\text{RW}}^{tot} = 3$.

the adjacency matrix. In terms of time complexity, computing the Shannon entropy scales as $\mathcal{O}(n)$. The calculation of MERW has instead an $\mathcal{O}(n^3)$ complexity due to the eigendecomposition required to compute the spectrum of the adjacency matrix.

It is worth noting that, in preliminary experiments, we have additionally investigated objective functions related to the Kolmogorov complexity. Also known as algorithmic complexity, this measure does not suffer from distributional dependencies [32]. As the Kolmogorov complexity is theoretically incomputable [9], we used graph compression algorithms such as *bzip-2* [11] and Block Decomposition Methods [53] to approximate the Kolmogorov complexity. However, as these approximations depend on the representation of the graph such as the adjacency matrix, one has to consider many permutations of the graph representation. Compressing the representation for a sufficient number of permutations becomes infeasible even for small graphs. While the MERW objective function is also derived from the adjacency matrix through its largest eigenvalue, it does not suffer from this artifact as the spectrum of the adjacency matrix is invariant to permutations.

## 5.2 Simulating and evaluating attacker behavior

Given an initial connected and undirected graph $G_0 = (\mathcal{V}, \mathcal{E}_0)$, we model the attacker as having entered the network through an arbitrary node $u \in \mathcal{V}$, and having built a *local map* $\mathcal{M}_0^u = (\mathcal{V}^u, \mathcal{E}_0^u)$ around this entry point, where $\mathcal{V}^v \subset \mathcal{V}$ is the set of nodes and $\mathcal{E}_0^u \subset \mathcal{E}_0$ is the set of edges in the map. The rewiring procedure transforms the initial graph $G_0 = (\mathcal{V}, \mathcal{E}_0)$ to the graph $G_* = (\mathcal{V}, \mathcal{E}_*)$, yielding the new local map $\mathcal{M}_*^u = (\mathcal{V}^u, \mathcal{E}_*^u)$ that is unknown to the attacker. Our goal is to evaluate the effectiveness of the reconfiguration by measuring how "stale" the prior information of the attacker has become in comparison to the new map: if the attacker struggles to find its targets in the updated topology, the rewiring has succeeded.

Let $\overline{\mathcal{V}^u}$ denote the set of nodes in the new local map $\mathcal{M}_*^u$ that are unreachable through *at least one* trajectory composed of original edges $E_0^u$ in the old map. For each newly unreachable node $v_i$, we measure the cost $\mathcal{C}_{\text{RW}}(v_i)$ of finding it with a *forward random walk*, in which the random walker only returns to the previous node if the current node has no other outgoing links. Every time the random walker encounters a link that is (i) not included in $E_0^u$ and (ii) not yet encountered during the random walk, the cost increases by one. This simulates the cost of having to explore the new graph topology due to the reconfigurations that were introduced. Finally, we let $\mathcal{C}_{\text{RW}}^{tot} = \sum_{v_i \in \overline{\mathcal{V}^u}} \mathcal{C}_{\text{RW}}(v_i)$ denote the sum of the costs for all newly unreachable nodes, which is our metric for the effectiveness of a rewiring strategy. An illustrative example of a forward random walk and cost evaluation is shown in Figure 2, and a formal description is presented in Algorithm 1 in Appendix A to aid reproducibility.

# 6 Experiments

## 6.1 Experimental setup

**Training and evaluation procedure.** Our agent is trained on synthetic graphs of size $n = 30$ that are generated using the graph models listed below. Every agent has a budget $b$, defined as a percentage of the total edges $m$ in the graph. This definition is based on the normalization using the total number of edges and enables consistent comparisons over different graph sizes and topologies. Where not specified otherwise, we use $b = 15\%$. When performing the attacker simulations, the

initial local map contains the subgraph induced by all nodes that are 2 hops away from the entry point, which is sampled without replacement from the node set. Training occurs separately for each graph model and objective $\mathcal{F}$ on a set of graphs $\mathcal{G}_{\text{train}}$ of size $|\mathcal{G}_{\text{train}}| = 6 \cdot 10^2$. Every 10 training steps, we measure the performance on a disjoint validation set $\mathcal{G}_{\text{validation}}$ of size $|\mathcal{G}_{\text{validation}}| = 2 \cdot 10^2$. We perform reconfiguration operations on a test set $\mathcal{G}_{\text{test}}$ of size $|\mathcal{G}_{\text{test}}| = 10^2$. To account for stochasticity, we train our models with 10 different seeds and present mean and confidence intervals accordingly. Further details about the experimental procedure (e.g., hyperparameter optimization) can be found in the Appendix A.

**Synthetic graphs.** We evaluate the approaches on graphs generated by the following models:

*Barabási–Albert (BA)*: A preferential attachment model where nodes joining the network are linked to $M$ nodes [4]. We consider values of $M_{ba} = 2$ and $M_{ba} = 1$ (abbreviated BA-2 and BA-1).

*Watts–Strogatz (WS)*: A model that starts with a ring lattice of nodes with degree $k$. Each edge is rewired to a random node with probability $p$, yielding characteristically small shortest path lengths [49]. We use $k = 4$ and $p = 0.1$.

*Erdős–Rényi (ER)*: A random graph model in which the existence of each edge is governed by a uniform probability $p$ [19]. We use $p = 0.15$.

**Real-world graphs.** We also consider the real-world Unified Host and Network (UHN) dataset [46], which is a subset of network and host events from an enterprise network. We transform this dataset into a graph by identifying the bidirectional links between hosts appearing in these records, obtaining a graph with $n = 461$ nodes and $m = 790$ edges. Further information about this processing can be found in Appendix A.

**Baselines.** We compare the entropy maximization method against two baselines: *Random*, which acts in the same MDP as the agent but chooses actions uniformly, and *Greedy*, which is a shallow one-step search over all rewirings from a given configuration. The latter selects the rewiring that provides the largest improvement in $\mathcal{F}$. Besides Random and Greedy, we compare our intrusion defense method to a third baseline named *MinConnectivity*. This baseline is a modification of the greedy heuristic introduced by [21] and aims to decrease the algebraic connectivity of a graph based on the Fiedler vector [20] $v$. It performs the rewiring by removing the existing edge $(i, j)$ with the largest contribution $(v_i - v_j)^2$ to the algebraic connectivity, and adding the edge $(j, k)$ with the smallest $(v_j - v_k)^2$. The motivation behind this baseline is that decreasing the connectivity of the graph would impede / slow down the navigation task of the intruder.

## 6.2 Entropy maximization results

We first consider the results for the maximization of the entropy-based objectives. The gains in entropy obtained by the methods on the held-out test set are shown in Table 1, while training curves are presented in Section 6.4. The results demonstrate that the approach discovers better reconfiguration strategies than random rewiring in all cases, and even the greedy search in one setting. Furthermore, we evaluate the out-of-distribution generalization properties of the learned models along two dimensions: varying the graph size $n \in [10, 300]$ and the budget $b$ as a percentage of existing edges

**Table 1:** Entropy gains on test graphs with $n = 30$ and a budget of $15\%$.

| $\mathcal{F}$ | $\mathcal{G}_{\text{test}}$ | DQN | Greedy | Random |
|---|---|---|---|---|
| $\Delta\mathcal{F}_{MERW}$ | BA-2 | $0.197_{\pm 0.002}$ | $0.225_{\pm 0.003}$ | $-0.019_{\pm 0.003}$ |
| | BA-1 | $0.167_{\pm 0.003}$ | $0.135_{\pm 0.003}$ | $-0.045_{\pm 0.004}$ |
| | ER | $0.182_{\pm 0.004}$ | $0.209_{\pm 0.012}$ | $-0.005_{\pm 0.003}$ |
| | WS | $0.233_{\pm 0.003}$ | $0.298_{\pm 0.002}$ | $0.035_{\pm 0.002}$ |
| $\Delta\mathcal{F}_{Shannon}$ | BA-2 | $0.541_{\pm 0.009}$ | $0.724_{\pm 0.015}$ | $0.252_{\pm 0.024}$ |
| | BA-1 | $0.167_{\pm 0.008}$ | $0.242_{\pm 0.012}$ | $0.084_{\pm 0.015}$ |
| | ER | $0.101_{\pm 0.012}$ | $0.400_{\pm 0.023}$ | $-0.022_{\pm 0.018}$ |
| | WS | $0.926_{\pm 0.016}$ | $1.116_{\pm 0.022}$ | $0.567_{\pm 0.036}$ |

$\in \{5, 10, 15, 20, 25\}$. The results for this experiment are shown in Figure 3. We do not report results for the Greedy solution since it is characterized by very poor scalability and, therefore, it is not practical. We find that, with the exception of the (BA, $\mathcal{F}_{Shannon}$) combination, the learned models generalize well to graphs substantially larger in size as well as varying rewiring budgets.

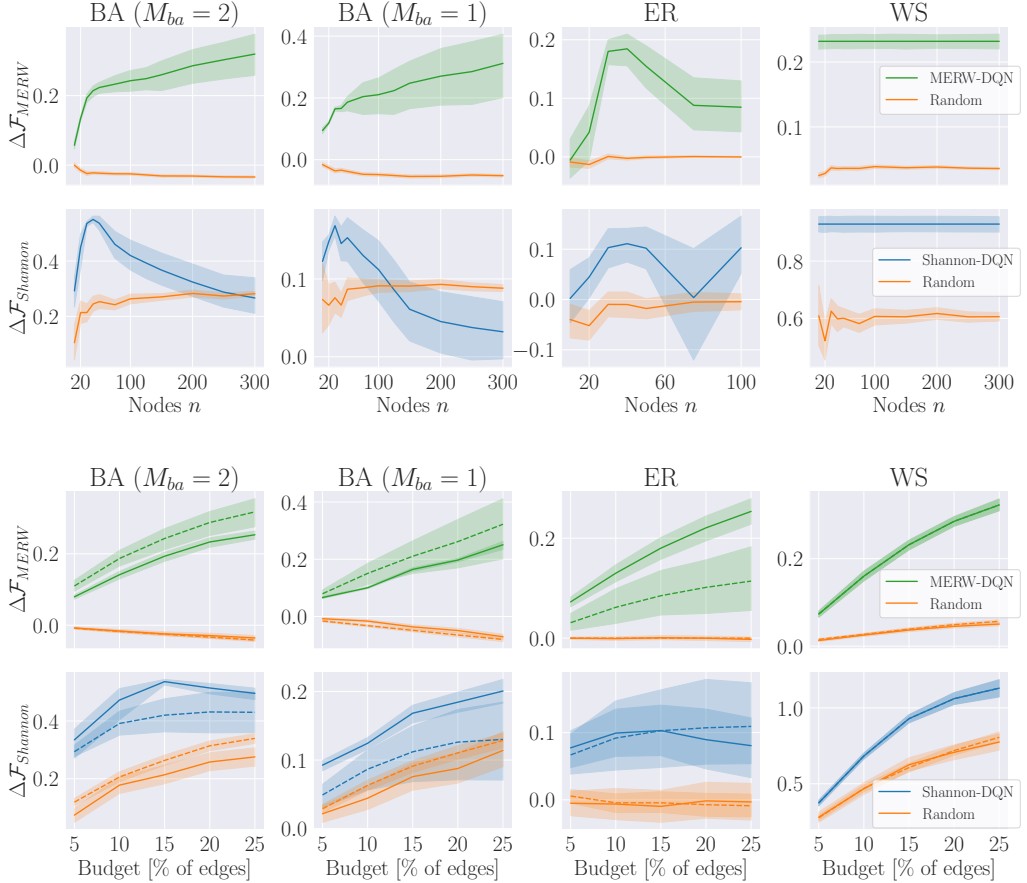

**Figure 3:** Evaluation of the out-of-distribution generalization performance (higher is better) of the learned entropy maximization models as a function of graph size (top) and budget size (bottom). All models are trained on graphs with $n = 30$. In the top figure, the applied budget is $15\%$. In the bottom figure, the solid and dotted lines represent graphs with $n = 30$ and $n = 100$ respectively. Note the different x-axes used for ER graphs due to their high edge density.

### 6.3 Evaluating the reconfiguration impact

We next evaluate the performance of the learned models for entropy maximization on the downstream task of disrupting the navigation of the graph by the attacker.

**Synthetic graphs.** The results for synthetic graphs are shown in Figure 4 in an out-of-distribution setting as a function of graph size, a regime in which the Greedy baseline is too expensive to scale. We find that the best proxy metric varies with the class of synthetic graphs – Shannon entropy performs better for BA graphs, MERW is better for ER, and performance is similar for WS. Strong out-of-distribution generalization performance is observed for 3 out of 4 synthetic graph models. The results also show that, in the case of WS graphs, even if we observe high performance in relation to the metric (as shown in

**Table 2:** Total random walk cost of models applied to the real-world UHN graph ($n = 461, m = 790, b = 15\%$).

|  | $\mathcal{F}$ |  | $\mathcal{C}_{\mathrm{RW}}^{tot}/n\,(\uparrow)$ |
|---|---|---|---|
| DQN | $\mathcal{F}_{MERW}$ | BA-2 | $3.087_{\pm 0.225}$ |
|  |  | BA-1 | $1.294_{\pm 0.185}$ |
|  |  | ER | $2.887_{\pm 0.335}$ |
|  |  | WS | $\mathbf{4.888}_{\pm 0.568}$ |
|  | $\mathcal{F}_{Shannon}$ | BA-2 | $3.774_{\pm 0.445}$ |
|  |  | BA-1 | $\mathbf{4.660}_{\pm 0.461}$ |
|  |  | ER | $3.891_{\pm 0.559}$ |
|  |  | WS | $3.555_{\pm 0.318}$ |
| Random | — | — | $2.071_{\pm 0.289}$ |
| MinConnectivity | — | — | $2.086_{\pm 0.671}$ |
| Greedy | — | — | $\infty$ |

Figure 3), the objective is not a suitable proxy for the downstream task in an out-of-distribution setting since the random walk cost decays rapidly. This might be explained by the fact that the graph topology is derived through a rewiring process of cliques of nodes of a given size. Finally,

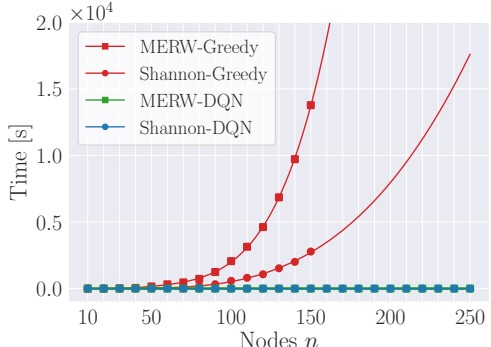

**Figure 4:** Evaluation of the learned rewiring strategies for entropy maximization on the downstream task of disrupting attacker navigation. All models are trained on graphs with $n = 30$ and have a budget $b$ of $15\%$. The random walk cost $\mathcal{C}_{\text{RW}}^{tot}$ (higher is better) is normalized by $n$ for meaningful comparisons. Note the different x-axis used for ER graphs due to their high edge density.

both DQN agents outperform Random and MinConnectivity on BA-2 and ER graphs. In the BA-1 setting, the Shannon DQN outperforms the baselines on BA-1 graphs in the small-$n$ domain, while MinConnectivity is clearly superior for large $n$. The baseline likely converts the sparse graph into a long string (which has very low connectivity), resulting in large random walk costs. In contrast, DQN aims to maximize entropy and therefore avoids strings, which have low entropy due to the monotonic node degrees of the sequence.

**Real-world graphs.** We also evaluate the models trained on synthetic graphs on the real-world graph constructed from the UHN dataset. Results are shown in Table 2. All but one of the trained models maintain a statistically significant random walk cost difference over the Random and MinConnectivity baselines. The best-performing models were trained on the (WS, $\mathcal{F}_{MERW}$) and (BA-1, $\mathcal{F}_{Shannon}$) combinations, obtaining total gains in random walk cost $\mathcal{C}_{\text{RW}}^{tot}$ of $136\%$ and $125\%$ respectively. The Greedy baseline is not applicable for a graph of this size.

## 6.4 Learning curves

Learning curves are shown in Figure 5, which captures the performance on the held-out validation set $\mathcal{G}_{\text{validation}}$. We note that in many cases (e.g., BA / $\mathcal{F}_{MERW}$) the performance averaged across all seeds is misleadingly low compared to the baselines, an artifact of the variability of the validation set performance. We also show the performance of the worst-performing seed (dotted) and best-performing seed (dashed) to clarify this.

## 6.5 Time complexity

To evidence the poor scalability of the Greedy baseline as discussed in Section 6.1, we perform an additional experiment that measures the wall clock time taken by the different approaches to complete a sequence of rewirings. Results are shown in Figure 6 for Barabási-Albert graphs ($M_{ba} = 2$) as a function of graph size. Beyond graphs of size $n = 150$, we extrapolate by fitting polynomials of degree 5 and 4 for $\mathcal{F}_{MERW}$ and $\mathcal{F}_{Shannon}$ respectively.

The time needed for evaluating the Greedy baseline increases rapidly as the size of the graph grows, while the post-training DQN is very efficient from a computational point of view. Hence, it is not feasible to use the Greedy baseline beyond very small graphs, but it serves as a useful comparison point.

**Figure 6:** Wall clock time needed to complete a sequence of rewirings by the Greedy and DQN methods on Barabási-Albert graphs ($M_{ba} = 2$) with a rewiring budget of $15\%$.

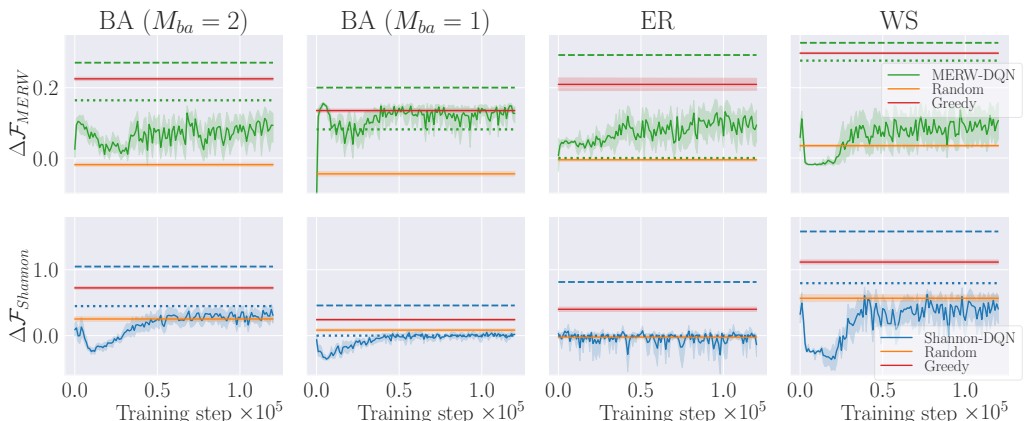

**Figure 5:** MERW (upper half) and Shannon entropy (lower half) increase on the held-out validation set $\mathcal{G}_{\text{validation}}$ during training of the DQN algorithm. The dotted and dashed lines for the DQN algorithm represent the worst-performing and best-performing seeds respectively. Random and Greedy rewiring performance are shown for comparison. Graphs are of size $n = 30$ and the rewiring budget is $15\%$ of the number of existing edges.

## 7 Conclusion

**Summary.** In this work, we have addressed the problem of graph reconfiguration for the optimization of a given property of a networked system, a computationally challenging problem given the generally large decision space. We have then formulated it as a Markov Decision Process that treats rewirings as sequential and proposed an approach based on deep reinforcement learning and graph neural networks for efficient learning of network reconfigurations. As a case study, we have applied the proposed method to a cybersecurity scenario in which the task is to disrupt the navigation of potential intruders in a computer network. We have assumed that the goal of the intruder is to navigate the network given some knowledge about its topology. In order to disrupt the attack, we have designed a mechanism for increasing the level of surprise of the network through entropy maximization by means of network rewiring. More specifically, in terms of the objective of the optimization process, we have considered two entropy metrics that quantify the predictability of the network topology, and demonstrated that our method generalizes well on unseen graphs with varying rewiring budgets and different numbers of nodes. We have also validated the effectiveness of the learned models for increasing path lengths towards targeted nodes. The proposed approach outperforms the considered baselines on both synthetic and real-world graphs.

**Limitations and future work.** An advantage of the proposed approach is that it does not require any knowledge of the exact position of the attacker as the traversal of the graph takes place. One may also consider a real-time scenario in which the network reconfiguration aims to "close off" the attacker given knowledge of their location, which may lead to a more efficient defense if such information is available. We have also adopted a simple model of attacker navigation (forward random walks). Different, more complex navigation strategies (e.g., targeting vulnerable machines) can also be considered. This knowledge might be integrated as part of the training process, for example by increasing the probability of rewiring of edges around these nodes through a corresponding reward structure (i.e., higher reward for protecting more sensitive nodes). More generally, we have identified an important application to cybersecurity, which might have a positive impact in safeguarding networks from malicious intrusions.

## Acknowledgements

This work was supported by The Alan Turing Institute under the UK EPSRC grant EP/N510129/1. The authors declare that they have no competing interests with respect to this work.

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

# A   Implementation and training details

**Codebase.**   The code for reproducing the results of this work is available in the Supplementary Material and in the online code repository at `https://github.com/ChristoffelDoorman/network-rewiring-rl`. The DQN implementation we use is bootstrapped from the RNet-DQN codebase[1] in [15], which itself is based on the RL-S2V[2] implementation from [13] and S2V GNN[3] from [12]. Our neural network architecture is implemented with the deep learning library PyTorch [38].

**Infrastructure and runtimes.**   Experiments were carried out on a cluster of 8 machines, each equipped with 2 Intel Xeon E5-2630 v3 processors and 128GB RAM. On this infrastructure, all experiments reported in this paper took approximately 8 days to complete.

**MDP parameters.**   To improve numerical stability we scale the reward signals in Equation 5 by $c_{\mathcal{F}} = 10^1$ for MERW-DQN and $c_{\mathcal{F}} = 10^2$ for Shannon-DQN. We set the disconnection penalty $\bar{r}_n = -10.0$. As we consider a finite horizon MDP, we set the discount factor $\gamma = 1$.

**Model architectures and hyperparameters.**   In all experiments the same neural network architectures and hyperparameters are used in the three stages of the rewiring procedure as described in Section 3. The final MLPs described in Equation 8 contain a hidden layer of 128 units and a single-unit output layer representing the estimated state-action value. Batch normalization [28] is applied to the input of the final layer.

We performed an initial hyperparameter grid search on BA-2 graphs over the following search space: the initial learning rate $\alpha_0 \in \{5, 10, 50\} \cdot 10^{-4}$ for MERW-DQN and $\alpha_0 \in \{1, 5, 10\} \cdot 10^{-4}$ for Shannon-DQN; the number of message-passing rounds $L \in \{3, 4\}$; the latent dimension of the graph embedding $\dim(\mu_i) \in \{32, 64, 128\}$. Due to

**Table 3:** Optimal initial learning rate $\alpha_0$, message passing rounds $L$ and graph embedding dimension $\dim(\mu_i)$ found by a hyperparameter search.

| DQN | $\mathcal{G}$ | $\alpha_0 \, [10^{-4}]$ | $L$ | $\dim(\mu_i)$ |
|---|---|---|---|---|
| $\mathcal{F}_{MERW}$ | BA-2 | 5 | 3 | 128 |
| | BA-1 | 5 | 6 | 128 |
| | ER | 5 | 4 | 128 |
| | WS | 10 | 6 | 128 |
| $\mathcal{F}_{Shannon}$ | BA-2 | 10 | 3 | 64 |
| | BA-1 | 5 | 6 | 64 |
| | ER | 1 | 4 | 64 |
| | WS | 10 | 6 | 64 |

computational budget constraints, for BA-1, ER and WS graphs, we only performed a hyperparameter search for for the initial learning rate $\alpha_0$ over the same values as for BA-2 graphs, while setting the number of message passing rounds equal to the graph diameter $L = D$ and bootstrapping the latent dimension from the hyperparameter search on BA-2 graphs. Table 3 presents an overview of the optimal values of the hyperparameters that were used for the results presented in the paper.

**Training details.**   We train the models for $120,000$ steps, and let the exploration parameter $\varepsilon$ decay linearly from $\varepsilon = 1.0$ to $\varepsilon = 0.1$ in the first $40,000$ training steps after which it is kept constant. The network parameters are initialized using Glorot initialization [22] and updated using the Adam optimizer [31]. We use a batch size of 50 graphs. The replay memory contains 12,000 instances and replaces the oldest entry when adding a new transition. The target network parameters are updated every 50 training steps.

**Graphs.**   The real-world UHN dataset [46] contains network events on day 2 of approximately 90 days of network events collected from the Los Alamos National Laboratory enterprise network and is pre-processed as follows: firstly, we build a directional graph where nodes represent unique hosts in the data set and construct directional links from the events between the hosts. Secondly, we filter the graph by removing all unidirectional links and transform the graph to be undirected, only keeping the largest connected component. Thirdly, we exclude nodes that only have many single-degree neighbors, such as email servers, and furthermore only retain nodes with degrees $\leq 80$. The graph obtained by this procedure is illustrated in Figure 7. We additionally note that, in all downstream experiments, graphs that are disconnected after rewiring are not considered in any of the evaluations.

**Reconfiguration impact evaluation.**   The algorithm we use for measuring the random walk cost $\mathcal{C}_{RW}$ induced by a sequence of rewirings is shown in Algorithm 1. We sample without replacement

---

[1] `https://github.com/VictorDarvariu/graph-construction-rl`
[2] `https://github.com/Hanjun-Dai/graph_adversarial_attack`
[3] `https://github.com/Hanjun-Dai/pytorch_structure2vec`

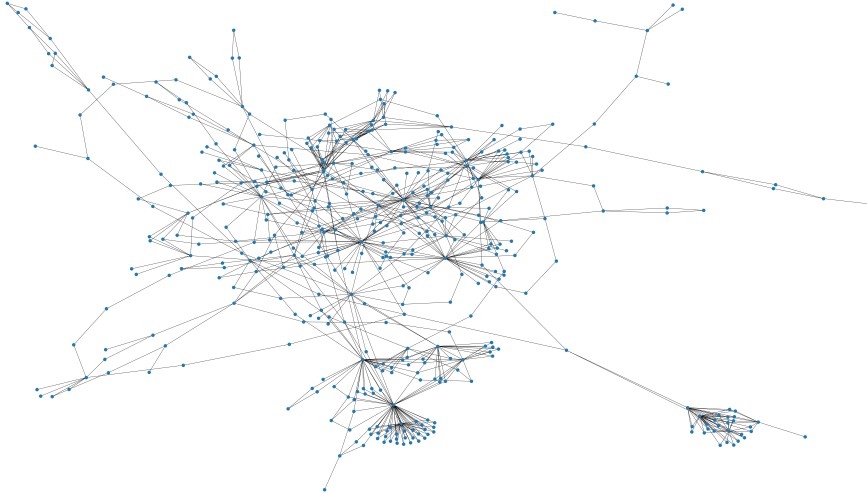

**Figure 7:** The graph derived from the Unified Host and Network (UHN) data set. It contains $n = 461$ nodes, $m = 790$ edges, and has a diameter $D = 18$.

$N_{\text{synthetic}} = \min\{n, 30\}$ and $N_{\text{UHN}} = n$ entry nodes for synthetic graphs and the UHN graph, respectively. After rewiring, we find the nodes that have become unreachable through at least one trajectory composed of the edges of the old map. We then perform a single random walk per missing target node as described in Section 5.2 and Algorithm 1.

---

**Algorithm 1:** Random walk cost evaluation

**Data:** $G_*(\mathcal{V}, \mathcal{E}_*), u, v_i \in \mathcal{V}, E_0^u \subset \mathcal{E}_0;$     // $u$, $v_i$ are entry, target node resp.

$\mathcal{C}_{\text{RW}} \leftarrow 0;$

$\mathcal{E}_{visited} \leftarrow (v_j, v_k) \in E_0^u \; \forall j, k;$

$v_{t-1}, v_t \leftarrow u \in \mathcal{V};$     // $v_{t-1}$, $v_t$ are previous, current position resp.

$v_{t+1} \leftarrow \mathcal{U}(\mathcal{N}_u);$     // $v_{t+1}$ is next position

**while** $v_{t+1} \neq v_i$ **do**

    $e_t \leftarrow (v_t, v_{t+1});$

    **if** $e_t \notin \mathcal{E}_{visited}$ **then**

        $\mathcal{C}_{\text{RW}} \leftarrow \mathcal{C}_{\text{RW}} + 1;$

        add $e_t$ to $\mathcal{E}_{visited};$

    **end**

    **if** $k_{v_{t+1}} = 1$ **then**

        $v_{t-1} \leftarrow v_{t+1};$     // reverse random walk if dead end

    **else**

        $v_{t-1} \leftarrow v_t;$

        $v_t \leftarrow v_{t+1};$

    **end**

    $v_{t+1} \leftarrow \mathcal{U}(\mathcal{N}_{v_t} \setminus v_{t-1});$     // choose next node randomly

**end**

$e_t \leftarrow (v_t, v_{t+1})$ **if** $e_t \notin \mathcal{E}_{visited}$ **then**

    $\mathcal{C}_{\text{RW}} \leftarrow \mathcal{C}_{\text{RW}} + 1;$

**end**

---

