# OpenReview forum: "Dynamic Network Reconfiguration for Entropy Maximization using Deep Reinforcement Learning"
_logconference.io/LOG/2022/Conference — LoG 2022 Poster_

### Official Review · Reviewer_iuJR · 2022-10-17

**Overall Score:** 5
**Confidence:** 4

**Review:**

## Overview:
The authors consider the problem of optimizing structural properties of a graph by rewiring edges. They propose a reinforcement learning algorithm combining deep Q-networks (DQN) with graph neural networks (GNNs). They evaluate their method on a cybersecurity task, which they show is equivalent to a certain entropy maximization problem. Their method outperforms random rewiring of edges but in general does not beat a greedy strategy. The authors show that (unlike the greedy strategy), their method scales well with graph size. Additionally, they show that their method generalizes well to unseen instances.

## Strengths:
The authors consider an important problem that has a variety of important applications. They propose a generic algorithm that could be applied to a broad range of applications, and in principle can be used to maximize an arbitrary objective. They clearly describe their method and evaluate it on a variety of synthetic and real-world graphs. Overall, the paper is very well-written and easy to follow.

## Weaknesses:

### Baselines
The authors consider two baselines: a random one, and a greedy one. Given the task, the random baseline is a relative low bar to clear; the proposed method does not outperform any other baseline method. Their method consistently underperforms in comparison with the greedy strategy. I believe a stronger evaluation is needed for the empirical results to be convincing. In particular, one can either 1) identify stronger baselines for the studied cybersecurity task which the proposed method beats, or 2) consider a more well-studied task like e.g. maximizing the algebraic connectivity of a graph under rewiring (instead of edge additions), and adapt some of the existing heuristic methods (e.g. [3]) to the rewiring setting as baselines.

### Comparison with [2]
The authors of [2] consider the problem of optimizing a given objective on a graph via "constructing", i.e. adding edges. They also formalize the problem as an Markov decision process (MDP) and use of GNNs and the DQN algorithm. Even more, the choice of the reward functions, among other design choices, is completely analogous for [2] and this work. Given the similarity of both the problems considered, and the methods employed, I strongly believe a more explicit discussion of the merits of the current work is needed.

### Misc
No code is provided to reproduce the results.

## Questions:
- [1] defines entropy for an *ensemble* of networks, using the probability (over members of the ensemble) of a given edge being present. The authors cite [1] as defining "Shannon entropy of the degree distribution" and claim that it "captures the idea that graphs with heterogeneous degrees are less predictable than regular graphs". I do not see the how the equivalence that the authors claim follows. In fact, I am unsure of how the model of [1] is even applied to single graphs. My guess before checking out [1] was that the authors are treating the list of node degrees as a distribution and report its entropy.
- What do the authors mean by "We precede the edge removal step by edge addition to suppress potential disconnections of the graph"? Regardless of the order of operations (addition/deletion), rewiring can make a connected graph disconnected.
- Looking at the times reported in Figure 5, it seems reasonable to report results for the greedy algorithm for the bottom part of Figure 3 (each greedy run should take ~2000 seconds or ~30 minutes). Does the gap between greedy and the proposed DQN method increase as $n$ grows?

## Recommendation:
The existence of (virtually) the same method in the literature, solving a (very) closely related problem -- edge *additions* instead of *rewiring* to optimize a certain structural property in a graph -- together with the relatively weak empirical results, outweigh the clarity of exposition. Thus, I recommend rejection at this stage. I am willing to change my score if a more convincing empirical evaluation in a scenario that clearly differentiates additions/rewiring is presented.

----
# Post-rebuttal

I appreciate the authors' efforts to improve the paper.

I am rather confused by the additional baseline added to the experiments.
1. The authors repeatedly claim that the graph rewiring problem is significantly different from the construction one and one cannot simply re-use/easily adapt existing algorithms/heuristics from the construction setting to the rewiring setting. While I do not want to minimize the authors' efforts, the adaptation of [3], deemed MinConnectivity in the paper, is rather straightforward. It is completely non-obvious to me why adapting other construction algorithms, including [2], is a non-trivial task.
2. The authors state "decreasing the connectivity of the graph would impede / slow down the navigation task of the intruder" as a motivation for using MinConnectivity as a defense. It is unclear to me whether this is a fair comparison, since MinConnectivity, unlike the proposed method, does not explicitly try to maximize entropy (and yet performs competitively).

I appreciate that the authors are open-sourcing their code. I am also satisfied with the response of the authors to my other questions/comments.

Overall, based on my comments above, I am keeping my score.


[1] Anand, Kartik, and Ginestra Bianconi. "Entropy measures for networks: Toward an information theory of complex topologies." Physical Review E 80.4 (2009): 045102.

[2] Darvariu, Victor-Alexandru, Stephen Hailes, and Mirco Musolesi. "Goal-directed graph construction using reinforcement learning." Proceedings of the Royal Society A 477.2254 (2021): 20210168.

[3] Ghosh, Arpita, and Stephen Boyd. "Growing well-connected graphs." Proceedings of the 45th IEEE Conference on Decision and Control. IEEE, 2006.

---

### Official Review · Reviewer_UxjA · 2022-10-18

**Overall Score:** 6
**Confidence:** 4

**Review:**

Summary:
1.	This paper studies the problem of network rewiring---in which edges are reconfigured while their total number remains constant---for optimizing a specified structural property, which has not been studied in the literature. The authors formulate this task as a Markov Decision Process (MDP) and then propose an approach based on Deep Q-Network (DQN) and graph neural networks (GNNs) for rewiring networks.
2.	The authors evaluate the proposed method in the context of defense against network intrusion by an attacker. Specifically, they first learn rewiring strategies to maximize an objective function, and then test the learnt strategies in simulation environments. Experiments demonstrate the effectiveness of the proposed approach.

Overall, I recommend a weak reject. The reasons are as follows.
Strengths:
1.	This paper studies a new problem, i.e., network rewiring, which has not been studied in the literature.
2.	The proposed method is easy and the presentation of the approach is clear.
3.	The authors consider a realistic cybersecurity case study to evaluate their method. They first formulate the task as an entropy maximization problem, and then perform a simulation to demonstrate the effectiveness of their proposed method.

Weaknesses:
1.	Many existing works have used reinforcement learning for graph reconfiguration such as [1]. Though these approaches are not specifically designed for network rewiring, the problems are substantially similar (otherwise the authors may want to highlight the differences and the new technical difficulties). Therefore, the technical contribution is minor.
2.	In the cybersecurity case study, the authors formulate the task as an entropy maximization problem. However, the explanation is unclear.
3.	The compared baselines (Greedy, Random) are not strong enough. The authors may want to add more baselines that are applicable in the realistic cybersecurity scenario, e.g., the construction-based methods such as [1].

Questions:
1.	This paper focuses on the network rewiring problem, while many existing methods have been proposed for graph reconfiguration. It seems that these methods are also applicable for network rewiring by easily limiting the number of edges. Is this the case? Why is network rewiring an important topic? The authors may want to further address my concerns and I will be willing to raise my score.
2.	Why can’t graph construction methods work in the cybersecurity scenario?
3.	Does the initial graph $G_0$ affect the final rewired graph? If not, can we simply implement network rewiring by using existing graph construction methods to add edges?
4.	The authors simplify the intrusion defense task as an entropy maximization problem. Is this a common setting in the cybersecurity scenario? The motivation of “increasing the surprise of the network” is unclear.
5.	How to determine the final timestep $T$ (in Line 140)?

[1] Victor-Alexandru Darvariu, Stephen Hailes, and Mirco Musolesi. Goal-directed graph construction using reinforcement learning. Proceedings of the Royal Society A: Mathematical, Physical and Engineering Sciences, 477(2254), 2021.

---

### Official Review · Reviewer_uSpk · 2022-10-22

**Overall Score:** 6
**Confidence:** 4

**Review:**

Description:
The authors in this paper propose a Deep Q-Network (DQN) algorithm and GNNs that can efficiently learn  strategies for rewiring networks (network reconfiguration) for optimizing a specified structural property. They cast this task as a Markov Decision Process (MDP), in which a decision maker has a limited budget for rewiring that is performed sequentially. They evaluate their method in a realistic cybersecurity scenario (example of moving target defense (MTD)) where an attacker with partial knowledge of the underlying network topology has entered a computer network aiming to reach a particular node. The goal here is to learn a rewiring process for modifying the structure of the graph preventing the attacker from reaching the target node while keeping the network still functional.  This can be viewed as an entropy maximization problem while two measures of network entropy i)Shannon entropy and ii)Maximal Entropy Random Walk (MERW) are used as proxy measurements of the difficulty of the task. Eventually, they compare the performance of the proposed method with Greedy and Random rewiring approaches as baselines on both synthetic and real-world graphs.

Strength:
The idea of incorporating DQN and GNN to efficiently learn network reconfiguration was interesting. Also evaluating the method in a cybersecurity case study and using different entropy measurements in the optimization problem was an intriguing way to perform the experiments. Rather a good amount of experiments are done by the authors to validate their method. Optimizing a global structural property through rewiring instead of constructing a graph through edge addition was a smart idea.

Weakness: However I still have a couple of concerns about the proposed method and experiments.

1. My main concern is regarding the comparisons. Although the authors pick 2 methods (Greedy and Random) as baseline methods, due to time complexity, the Greedy algorithm is not scalable to very large networks and comparison with the proposed method is not feasible. So, in this case, since there have been lots of interest in this line of work in the past decade, I was wondering if there is no state-of-the-art baseline that the authors can compare their method.
2. My second concern is the size of the utilized network. Both synthetic and real datasets are rather small. I am curious to see the performance of the proposed method in much larger networks. (e.g. n ~10000).
3. To evaluate the model on the real graph why is the model which is trained on the synthetic graphs used? Why not train it on the real graph? And do synthetic graphs which are used for training, have similar structural properties to the real network?

Minor comments:
1. In Table 1 what is the value of the budget? The budget “b” is considered as a percentage of existing edges ∈ {5, 10, 15, 20, 25}, however, it is not clear how the authors compared different methods for different values of “b”.
To conclude, I found the proposed method interesting and timely. However, I think the paper can still significantly improve in experiments specifically by conveying them on networks with larger sizes and using state-of-the-art baseline methods.

---

### Official Review · Reviewer_213D · 2022-10-29

**Overall Score:** 6
**Confidence:** 3

**Review:**

**Summary**

This paper studies the problem of graph/network reconfiguration in order to optimize a given quantifiable objective, which is an important problem in network theory. The authors formulate this problem as global graph structural property maximization with an MDP. Then they propose to combine GNN and DQN to solve this optimization problem in a deep reinforcement learning style. Two entropy metrics that quantity the predictability of the network topology is considered in the objective of the optimization process.
Experiments on both synthetic and real-world graphs validate the effectiveness of the proposed approach in comparison with the two baselines.

**Pros:**
* The idea of combining GNNs and RL to solve the problem in network theory such as graph reconfiguration has merits, which is a relatively rarely explored research area.
* Experiments are carefully designed and are conducted on both synthetic and real-world graph datasets. The case study is interesting and helps to understand the practical importance of the studied problem.
* The paper is generally well-written and almost clear everywhere. Note there is a grammar error in Line 308 Sec. 7 Conclusion, we have then have -> we have then

**Cons:**
* The connection between the network reconfiguration problem and adversarial attack on graph-structured data with edge rewriting such as [1] needs further clarification in my opinion. As the authors discussed in Sec. related work, it seems the difference mainly lies in the objective function as a network property or classification task. Could these methods be employed in network reconfiguration problems too? This is because the two chosen baselines are both simple and intuitive methods, it would be better to modify some of these very related works such as [1] or RLS2V that also use RL as baselines to further demonstrate the effectiveness of the designs of key components in the proposed method.
* Meanwhile, this is interesting if we can use one of the two objective functions as a surrogate model to generate edge rewiring samples and find out whether these samples can be transferred to the other setting of the task. I know this is out of the scope of this paper and I just bring it up as a further discussion here.
* It would be better to discuss the size of the decision space before and after the carefully designed MDP process in details. The complexity of naïvely checking all possible edges within a graph is $\mathcal{O}(n^2)$ from my understanding, what is this complexity of the MDP framework with the separate timestep design in the proposed method?
* For experiments, both the synthetic and real-world graphs seem to be small, so the proposed method's scalability is not demonstrated. Is this due to the setting of the network reconfiguration problem itself? If so, the authors are suggested to discuss this setting explicitly for further clarification.

[1] Attacking Graph Convolutional Networks via Rewiring, ICLR 2020

**Questions during the rebuttal period:**

Please address and clarify the cons above

**Reasons for score:**

Overall, I lean into acceptance more. I like the idea of bringing newly proposed techniques to the traditional network theory problems. My concerns mainly fall into the experimental settings and scalability of the proposed method. Hopefully, the authors can address my concern in the rebuttal period.

---

### Meta-Review · Area_Chair_6UH3 · 2022-11-09

**Confidence:** 4
**Recommendation:** Accept

**Meta Review:**

This work proposes an RL-based solution to an interesting problem of network reconfiguration. The concerns from reviewers concentrated on missing baselines and the technical difficulty in extending previous works to this new setting. I think based on the response from the authors, the concerns are mostly addressed. So, I recommend acceptance. For the final version, I strongly suggest the authors plug in more recent and challenging baselines, e.g., [1] as one reviewer suggested. Also, I am curious about the relationship between the proposed network reconfiguration method to handle network intrusion and the graph information bottleneck approach [2] to deal with adversarial attacks. The latter one also asks to learn to perturb the network to make the model more stable.  It would be good if the authors may include some discussion along this line in the final version.


[1] Attacking Graph Convolutional Networks via Rewiring, ICLR 2020.
[2] Graph information bottleneck, Wu et al. NeurIPS 2020.

---

### Decision · Program_Chairs · 2022-11-23

Accept (Poster)